# Rates of convergence for nearest neighbor classification

**Kamalika Chaudhuri**
Computer Science and Engineering
University of California, San Diego
kamalika@cs.ucsd.edu

**Sanjoy Dasgupta**
Computer Science and Engineering
University of California, San Diego
dasgupta@cs.ucsd.edu

## Abstract

We analyze the behavior of nearest neighbor classification in metric spaces and provide finite-sample, distribution-dependent rates of convergence under minimal assumptions. These are more general than existing bounds, and enable us, as a by-product, to establish the universal consistency of nearest neighbor in a broader range of data spaces than was previously known. We illustrate our upper and lower bounds by introducing a new smoothness class customized for nearest neighbor classification. We find, for instance, that under the Tsybakov margin condition the convergence rate of nearest neighbor matches recently established lower bounds for nonparametric classification.

## 1 Introduction

In this paper, we deal with binary prediction in metric spaces. A classification problem is defined by a metric space $(\mathcal{X}, \rho)$ from which instances are drawn, a space of possible labels $\mathcal{Y} = \{0, 1\}$, and a distribution $\mathbb{P}$ over $\mathcal{X} \times \mathcal{Y}$. The goal is to find a function $h : \mathcal{X} \to \mathcal{Y}$ that minimizes the probability of error on pairs $(X, Y)$ drawn from $\mathbb{P}$; this error rate is the *risk* $R(h) = \mathbb{P}(h(X) \neq Y)$. The best such function is easy to specify: if we let $\mu$ denote the marginal distribution of $X$ and $\eta$ the conditional probability $\eta(x) = \mathbb{P}(Y = 1 | X = x)$, then the predictor $1(\eta(x) \geq 1/2)$ achieves the minimum possible risk, $R^* = \mathbb{E}_X[\min(\eta(X), 1 - \eta(X))]$. The trouble is that $\mathbb{P}$ is unknown and thus a prediction rule must instead be based only on a finite sample of points $(X_1, Y_1), \ldots, (X_n, Y_n)$ drawn independently at random from $\mathbb{P}$.

Nearest neighbor (NN) classifiers are among the simplest prediction rules. The *1-NN classifier* assigns each point $x \in \mathcal{X}$ the label $Y_i$ of the closest point in $X_1, \ldots, X_n$ (breaking ties arbitrarily, say). For a positive integer $k$, the *k-NN classifier* assigns $x$ the majority label of the $k$ closest points in $X_1, \ldots, X_n$. In the latter case, it is common to let $k$ grow with $n$, in which case the sequence $(k_n : n \geq 1)$ defines a $k_n$-*NN classifier*.

The asymptotic consistency of nearest neighbor classification has been studied in detail, starting with the work of Fix and Hodges [7]. The risk of the NN classifier, henceforth denoted $R_n$, is a random variable that depends on the data set $(X_1, Y_1), \ldots, (X_n, Y_n)$; the usual order of business is to first determine the limiting behavior of the expected value $\mathbb{E}R_n$ and to then study stronger modes of convergence of $R_n$. Cover and Hart [2] studied the asymptotics of $\mathbb{E}R_n$ in general metric spaces, under the assumption that every $x$ in the support of $\mu$ is either a continuity point of $\eta$ or has $\mu(\{x\}) > 0$. For the 1-NN classifier, they found that $\mathbb{E}R_n \to \mathbb{E}_X[2\eta(X)(1 - \eta(X))] \leq 2R^*(1 - R^*)$; for $k_n$-NN with $k_n \uparrow \infty$ and $k_n/n \downarrow 0$, they found $\mathbb{E}R_n \to R^*$. For points in Euclidean space, a series of results starting with Stone [15] established consistency without any distributional assumptions. For $k_n$-NN in particular, $R_n \to R^*$ almost surely [5].

These consistency results place nearest neighbor methods in a favored category of nonparametric estimators. But for a fuller understanding it is important to also have rates of convergence. For

instance, part of the beauty of nearest neighbor is that it appears to adapt automatically to different distance scales in different regions of space. It would be helpful to have bounds that encapsulate this property.

Rates of convergence are also important in extending nearest neighbor classification to settings such as active learning, semisupervised learning, and domain adaptation, in which the training data is not a fully-labeled data set obtained by i.i.d. sampling from the future test distribution. For instance, in active learning, the starting point is a set of unlabeled points $X_1, \ldots, X_n$, and the learner requests the labels of just a few of these, chosen adaptively to be as informative as possible about $\eta$. There are many natural schemes for deciding which points to label: for instance, one could repeatedly pick the point furthest away from the labeled points so far, or one could pick the point whose $k$ nearest labeled neighbors have the largest disagreement among their labels. The asymptotics of such selective sampling schemes have been considered in earlier work [4], but ultimately the choice of scheme must depend upon finite-sample behavior. The starting point for understanding this behavior is to first obtain a characterization in the non-active setting.

## 1.1 Previous work on rates of convergence

There is a large body of work on convergence rates of nearest neighbor estimators. Here we outline some of the types of results that have been obtained, and give representative sources for each.

The earliest rates of convergence for nearest neighbor were *distribution-free*. Cover [3] studied the 1-NN classifier in the case $\mathcal{X} = \mathbb{R}$, under the assumption of class-conditional densities with uniformly-bounded third derivatives. He showed that $\mathbb{E}R_n$ converges at a rate of $O(1/n^2)$. Wagner [18] and later Fritz [8] also looked at 1-NN, but in higher dimension $\mathcal{X} = \mathbb{R}^d$. The latter obtained an asymptotic rate of convergence for $R_n$ under the milder assumption of non-atomic $\mu$ and lower semi-continuous class-conditional densities.

Distribution-free results are valuable, but do not characterize which properties of a distribution most influence the performance of nearest neighbor classification. More recent work has investigated different approaches to obtaining distribution-dependent bounds, in terms of the smoothness of the distribution.

A simple and popular smoothness parameter is the Holder constant. Kulkarni and Posner [12] obtained a fairly general result of this kind for 1-NN and $k_n$-NN. They assumed that for some constants $K$ and $\alpha$, and for all $x_1, x_2 \in \mathcal{X}$,

$$|\eta(x_1) - \eta(x_2)| \ \leq \ K\rho(x_1, x_2)^{2\alpha}.$$

They then gave bounds in terms of the Holder parameter $\alpha$ as well as covering numbers for the marginal distribution $\mu$. Gyorfi [9] looked at the case $\mathcal{X} = \mathbb{R}^d$, under the weaker assumption that for some function $K : \mathbb{R}^d \to \mathbb{R}$ and some $\alpha$, and for all $z \in \mathbb{R}^d$ and all $r > 0$,

$$\left| \eta(z) - \frac{1}{\mu(B(z,r))} \int_{B(z,r)} \eta(x)\mu(dx) \right| \leq K(z)r^\alpha.$$

The integral denotes the average $\eta$ value in a ball of radius $r$ centered at $z$; hence, this $\alpha$ is similar in spirit to the earlier Holder parameter, but does not require $\eta$ to be continuous. Gyorfi obtained asymptotic rates in terms of $\alpha$. Another generalization of standard smoothness conditions was proposed recently [17] in a "probabilistic Lipschitz" assumption, and in this setting rates were obtained for NN classification in bounded spaces $\mathcal{X} \subset \mathbb{R}^d$.

The literature leaves open several basic questions that have motivated the present paper. (1) Is it possible to give tight finite-sample bounds for NN classification in metric spaces, without any smoothness assumptions? What aspects of the distribution must be captured in such bounds? (2) Are there simple notions of smoothness that are especially well-suited to nearest neighbor? Roughly speaking, we consider a notion suitable if it is possible to sharply characterize the convergence rate of nearest neighbor for all distributions satisfying this notion. As we discuss further below, the Holder constant is lacking in this regard. (3) A recent trend in nonparametric classification has been to study rates of convergence under "margin conditions" such as that of Tsybakov. The best achievable rates under these conditions are now known: does nearest neighbor achieve these rates?

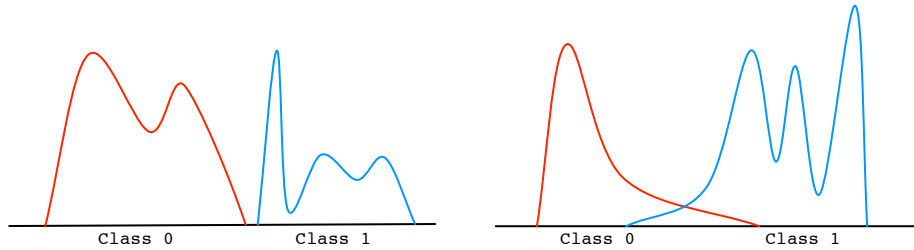

Figure 1: One-dimensional distributions. In each case, the class-conditional densities are shown.

## 1.2 Some illustrative examples

We now look at a couple of examples to get a sense of what properties of a distribution most critically affect the convergence rate of nearest neighbor. In each case, we study the $k$-NN classifier.

To start with, consider a distribution over $\mathcal{X} = \mathbb{R}$ in which the two classes ($Y = 0, 1$) have class-conditional densities $\mu_0$ and $\mu_1$. Assume that these two distributions have disjoint support, as on the left side of Figure 1. The $k$-NN classifier will make a mistake on a specific query $x$ only if $x$ is near the boundary between the two classes. To be precise, consider an interval around $x$ of probability mass $k/n$, that is, an interval $B = [x-r, x+r]$ with $\mu(B) = k/n$. Then the $k$ nearest neighbors will lie roughly in this interval, and there will likely be an error only if the interval contains a substantial portion of the wrong class. Whether or not $\eta$ is smooth, or the $\mu_i$ are smooth, is irrelevant.

In a general metric space, the $k$ nearest neighbors of any query point $x$ are likely to lie in a ball centered at $x$ of probability mass roughly $k/n$. Thus the central objects in analyzing $k$-NN are balls of mass $\approx k/n$ near the decision boundary, and it should be possible to give rates of convergence solely in terms of these.

Now let's turn to notions of smoothness. Figure 1, right, shows a variant of the previous example in which it is no longer the case that $\eta \in \{0, 1\}$. Although one of the class-conditional densities in the figure is highly non-smooth, this erratic behavior occurs far from the decision boundary and thus does not affect nearest neighbor performance. And in the vicinity of the boundary, what matters is not how much $\eta$ varies within intervals of any given radius $r$, but rather within intervals of probability mass $k/n$. Smoothness notions such as Lipschitz and Holder constants, which measure changes in $\eta$ with respect to $x$, are therefore not entirely suitable: what we need to measure are changes in $\eta$ with respect to the underlying marginal $\mu$ on $\mathcal{X}$.

## 1.3 Results of this paper

Let us return to our earlier setting of pairs $(X, Y)$, where $X$ takes values in a metric space $(\mathcal{X}, \rho)$ and has distribution $\mu$, while $Y \in \{0, 1\}$ has conditional probability function $\eta(x) = \Pr(Y = 1 | X = x)$. We obtain rates of convergence for $k$-NN by attempting to make precise the intuitions discussed above. This leads to a somewhat different style of analysis than has been used in earlier work.

Our main result is an upper bound on the misclassification rate of $k$-NN that holds for any sample size $n$ and for any metric space, with no distributional assumptions. The bound depends on a novel notion of the *effective boundary* for $k$-NN: for the moment, denote this set by $A_{n,k} \subset \mathcal{X}$.

- We show that with high probability over the training data, the misclassification rate of the $k$-NN classifier (with respect to the Bayes-optimal classifer) is bounded above by $\mu(A_{n,k})$ plus a small additional term that can be made arbitrarily small (Theorem 5).

- We lower-bound the misclassification rate using a related notion of effective boundary (Theorem 6).

- We identify a general condition under which, as $n$ and $k$ grow, $A_{n,k}$ approaches the actual decision boundary $\{x \mid \eta(x) = 1/2\}$. This yields universal consistency in a wider range of metric spaces than just $\mathbb{R}^d$ (Theorem 1), thus broadening our understanding of the asymptotics of nearest neighbor.

We then specialize our generalization bounds to smooth distributions.

- We introduce a novel smoothness condition that is tailored to nearest neighbor. We compare our upper and lower bounds under this kind of smoothness (Theorem 3).
- We obtain risk bounds under the margin condition of Tsybakov that match the best known results for nonparametric classification (Theorem 4).
- We look at additional specific cases of interest: when $\eta$ is bounded away from $1/2$, and the even more extreme scenario where $\eta \in \{0, 1\}$ (zero Bayes risk).

## 2 Definitions and results

Let $(\mathcal{X}, \rho)$ be any separable metric space. For any $x \in \mathcal{X}$, let
$$B^o(x, r) = \{x' \in \mathcal{X} \mid \rho(x, x') < r\} \quad \text{and} \quad B(x, r) = \{x' \in \mathcal{X} \mid \rho(x, x') \leq r\}$$
denote the open and closed balls, respectively, of radius $r$ centered at $x$.

Let $\mu$ be a Borel regular probability measure on this space (that is, open sets are measurable, and every set is contained in a Borel set of the same measure) from which *instances* $X$ are drawn. The *label* of an instance $X = x$ is $Y \in \{0, 1\}$ and is distributed according to the measurable conditional probability function $\eta : \mathcal{X} \to [0, 1]$ as follows: $\Pr(Y = 1 | X = x) = \eta(x)$.

Given a data set $S = ((X_1, Y_1), \ldots, (X_n, Y_n))$ and a query point $x \in \mathcal{X}$, we use the notation $X^{(i)}(x)$ to denote the $i$-th nearest neighbor of $x$ in the data set, and $Y^{(i)}(x)$ to denote its label. Distances are calculated with respect to the given metric $\rho$, and ties are broken by preferring points earlier in the sequence. The $k$-NN classifier is defined by

$$g_{n,k}(x) = \begin{cases} 1 & \text{if } Y^{(1)}(x) + \cdots + Y^{(k)}(x) \geq k/2 \\ 0 & \text{otherwise} \end{cases}$$

We analyze the performance of $g_{n,k}$ by comparing it with $g(x) = 1(\eta(x) \geq 1/2)$, the omniscient Bayes-optimal classifier. Specifically, we obtain bounds on $\Pr_X(g_{n,k}(X) \neq g(X))$ that hold with high probability over the choice of data $S$, for any $n$. It is worth noting that convergence results for nearest neighbor have traditionally studied the excess risk $R_{n,k} - R^*$, where $R_{n,k} = \Pr(Y \neq g_{n,k}(X))$. If we define the pointwise quantities

$$R_{n,k}(x) = \Pr(Y \neq g_{n,k}(x) | X = x)$$
$$R^*(x) = \min(\eta(x), 1 - \eta(x)),$$

for all $x \in \mathcal{X}$, we see that

$$R_{n,k}(x) - R^*(x) = |1 - 2\eta(x)| 1(g_{n,k}(x) \neq g(x)). \tag{1}$$

Taking expectation over $X$, we then have $R_{n,k} - R^* \leq \Pr_X(g_{n,k}(X) \neq g(X))$, and so we also obtain upper bounds on the excess risk.

The technical core of this paper is the finite-sample generalization bound of Theorem 5. We begin, however, by discussing some of its implications since these relate directly to common lines of inquiry in the statistical literature. All proofs appear in the appendix.

### 2.1 Universal consistency

A series of results, starting with [15], has shown that $k_n$-NN is strongly consistent ($R_n = R_{n,k_n} \to R^*$ almost surely) when $\mathcal{X}$ is a finite-dimensional Euclidean space and $\mu$ is a Borel measure. A consequence of the bounds we obtain in Theorem 5 is that this phenomenon holds quite a bit more generally. In fact, strong consistency holds in any metric measure space $(\mathcal{X}, \rho, \mu)$ for which the Lebesgue differentiation theorem is true: that is, spaces in which, for any bounded measurable $f$,

$$\lim_{r \downarrow 0} \frac{1}{\mu(B(x, r))} \int_{B(x,r)} f \, d\mu = f(x) \tag{2}$$

for almost all ($\mu$-a.e.) $x \in \mathcal{X}$.

For more details on this differentiation property, see [6, 2.9.8] and [10, 1.13]. It holds, for instance:

- When $(\mathcal{X}, \rho)$ is a finite-dimensional normed space [10, 1.15(a)].
- When $(\mathcal{X}, \rho, \mu)$ is *doubling* [10, 1.8], that is, when there exists a constant $C(\mu)$ such that $\mu(B(x, 2r)) \leq C(\mu)\mu(B(x, r))$ for every ball $B(x, r)$.
- When $\mu$ is an atomic measure on $\mathcal{X}$.

For the following theorem, recall that the risk of the $k_n$-NN classifier, $R_n = R_{n,k_n}$, is a function of the data set $(X_1, Y_1), \ldots, (X_n, Y_n)$.

**Theorem 1.** *Suppose metric measure space $(\mathcal{X}, \rho, \mu)$ satisfies differentiation condition (2). Pick a sequence of positive integers $(k_n)$, and for each $n$, let $R_n = R_{n,k_n}$ be the risk of the $k_n$-NN classifier $g_{n,k_n}$.*

1. *If $k_n \to \infty$ and $k_n/n \to 0$, then for all $\epsilon > 0$,*

$$\lim_{n\to\infty} \mathrm{Pr}_n(R_n - R^* > \epsilon) = 0.$$

   *Here $\mathrm{Pr}_n$ denotes probability over the data set $(X_1, Y_1), \ldots, (X_n, Y_n)$.*

2. *If in addition $k_n/(\log n) \to \infty$, then $R_n \to R^*$ almost surely.*

## 2.2 Smooth measures

Before stating our finite-sample bounds in full generality, we provide a glimpse of them under smooth probability distributions. We begin with a few definitions.

**The support of $\mu$.** The support of distribution $\mu$ is defined as

$$\mathrm{supp}(\mu) = \{x \in \mathcal{X} \mid \mu(B(x, r)) > 0 \text{ for all } r > 0\}.$$

It was shown by [2] that in separable metric spaces, $\mu(\mathrm{supp}(\mu)) = 1$. For the interested reader, we reproduce their brief proof in the appendix (Lemma 24).

**The conditional probability function for a set.** The conditional probability function $\eta$ is defined for points $x \in \mathcal{X}$, and can be extended to measurable sets $A \subset \mathcal{X}$ with $\mu(A) > 0$ as follows:

$$\eta(A) = \frac{1}{\mu(A)} \int_A \eta \, d\mu. \tag{3}$$

This is the probability that $Y = 1$ for a point $X$ chosen at random from the distribution $\mu$ restricted to set $A$. We exclusively consider sets $A$ of the form $B(x, r)$, in which case $\eta$ is defined whenever $x \in \mathrm{supp}(\mu)$.

### 2.2.1 Smoothness with respect to the marginal distribution

For the purposes of nearest neighbor, it makes sense to define a notion of smoothness with respect to the marginal distribution on instances. For $\alpha, L > 0$, we say the conditional probability function $\eta$ is $(\alpha, L)$-*smooth* in metric measure space $(\mathcal{X}, \rho, \mu)$ if for all $x \in \mathrm{supp}(\mu)$ and all $r > 0$,

$$|\eta(B(x, r)) - \eta(x)| \leq L \mu(B^o(x, r))^\alpha.$$

(As might be expected, we only need to apply this condition locally, so it is enough to restrict attention to balls of probability mass upto some constant $p_o$.) One feature of this notion is that it is scale-invariant: multiplying all distances by a fixed amount leaves $\alpha$ and $L$ unchanged. Likewise, if the distribution has several well-separated clusters, smoothness is unaffected by the distance-scales of the individual clusters.

It is common to analyze nonparametric classifiers under the assumption that $\mathcal{X} = \mathbb{R}^d$ and that $\eta$ is $\alpha_H$-*Holder continuous* for some $\alpha > 0$, that is,

$$|\eta(x) - \eta(x')| \leq L\|x - x'\|^{\alpha_H}$$

for some constant $L$. These bounds typically also require $\mu$ to have a density that is uniformly bounded (above and/or below). We now relate these standard assumptions to our notion of smoothness.

**Lemma 2.** *Suppose that $\mathcal{X} \subset \mathbb{R}^d$, and $\eta$ is $\alpha_H$-Holder continuous, and $\mu$ has a density with respect to Lebesgue measure that is $\geq \mu_{\min}$ on $\mathcal{X}$. Then there is a constant $L$ such that for any $x \in supp(\mu)$ and $r > 0$ with $B(x, r) \subset \mathcal{X}$, we have $|\eta(x) - \eta(B(x, r))| \leq L\mu(B^o(x, r))^{\alpha_H/d}$.*

(To remove the requirement that $B(x, r) \subset \mathcal{X}$, we would need the boundary of $\mathcal{X}$ to be well-behaved, for instance by requiring that $\mathcal{X}$ contains a constant fraction of every ball centered in it. This is a familiar assumption in nonparametric classification, including the seminal work of [1] that we discuss shortly.)

Our smoothness condition for nearest neighbor problems can thus be seen as a generalization of the usual Holder conditions. It applies in broader range of settings, for example for discrete $\mu$.

### 2.2.2 Generalization bounds for smooth measures

Under smoothness, our general finite-sample convergence rates (Theorems 5 and 6) take on an easily interpretable form. Recall that $g_{n,k}(x)$ is the $k$-NN classifier, while $g(x)$ is the Bayes-optimal prediction.

**Theorem 3.** *Suppose $\eta$ is $(\alpha, L)$-smooth in $(\mathcal{X}, \rho, \mu)$. The following hold for any $n$ and $k$.*

*(Upper bound on misclassification rate.) Pick any $\delta > 0$ and suppose that $k \geq 16 \ln(2/\delta)$. Then*

$$
\Pr_X(g_{n,k}(X) \neq g(X)) \leq \delta + \mu\left(\left\{x \in \mathcal{X} \;\middle|\; |\eta(x) - \frac{1}{2}| \leq \sqrt{\frac{1}{k}\ln\frac{2}{\delta}} + L\left(\frac{k}{2n}\right)^{\alpha}\right\}\right).
$$

*(Lower bound on misclassification rate.) Conversely, there is an absolute constant $c_o$ such that*

$$
\mathbb{E}_n \Pr_X(g_{n,k}(X) \neq g(X)) \geq c_o\mu\left(\left\{x \in \mathcal{X} \;\middle|\; \eta(x) \neq \frac{1}{2}, \; |\eta(x) - \frac{1}{2}| \leq \frac{1}{\sqrt{k}} - L\left(\frac{2k}{n}\right)^{\alpha}\right\}\right).
$$

*Here $\mathbb{E}_n$ is expectation over the data set.*

The optimal choice of $k$ is $\sim n^{2\alpha/(2\alpha+1)}$, and with this setting the upper and lower bounds are directly comparable: they are both of the form $\mu(\{x : |\eta(x) - 1/2| \leq \tilde{O}(k^{-1/2})\})$, the probability mass of a band of points around the decision boundary $\eta = 1/2$.

It is noteworthy that these upper and lower bounds have a pleasing resemblance for *every* distribution in the smoothness class. This is in contrast to the usual minimax style of analysis, in which a bound on an estimator's risk is described as "optimal" for a class of distributions if there exists even a single distribution in that class for which it is tight.

### 2.2.3 Margin bounds

An achievement of statistical theory in the past two decades has been *margin bounds*, which give fast rates of convergence for many classifiers when the underlying data distribution $\mathbb{P}$ (given by $\mu$ and $\eta$) satisfies a *large margin condition* stipulating, roughly, that $\eta$ moves gracefully away from $1/2$ near the decision boundary.

Following [13, 16, 1], for any $\beta \geq 0$, we say $\mathbb{P}$ satisfies the $\beta$-*margin condition* if there exists a constant $C > 0$ such that

$$
\mu\left(\left\{x \;\middle|\; |\eta(x) - \frac{1}{2}| \leq t\right\}\right) \leq Ct^{\beta}.
$$

Larger $\beta$ implies a larger margin. We now obtain bounds for the misclassification rate and the excess risk of $k$-NN under smoothness and margin conditions.

**Theorem 4.** *Suppose $\eta$ is $(\alpha, L)$-smooth in $(\mathcal{X}, \rho, \mu)$ and satisfies the $\beta$-margin condition (with constant $C$), for some $\alpha, \beta, L, C \geq 0$. In each of the two following statements, $k_o$ and $C_o$ are constants depending on $\alpha, \beta, L, C$.*

(a) *For any $0 < \delta < 1$, set $k = k_o n^{2\alpha/(2\alpha+1)}(\log(1/\delta))^{1/(2\alpha+1)}$. With probability at least $1 - \delta$ over the choice of training data,*

$$
\Pr_X(g_{n,k}(X) \neq g(X)) \leq \delta + C_o\left(\frac{\log(1/\delta)}{n}\right)^{\alpha\beta/(2\alpha+1)}.
$$

*(b) Set $k = k_o n^{2\alpha/(2\alpha+1)}$. Then $\mathbb{E}_n R_{n,k} - R^* \leq C_o n^{-\alpha(\beta+1)/(2\alpha+1)}$.*

It is instructive to compare these bounds with the best known rates for nonparametric classification under the margin assumption. The work of Audibert and Tsybakov [1] (Theorems 3.3 and 3.5) shows that when $(\mathcal{X}, \rho) = (\mathbb{R}^d, \|\cdot\|)$, and $\eta$ is $\alpha_H$-Holder continuous, and $\mu$ lies in the range $[\mu_{\min}, \mu_{\max}]$ for some $\mu_{\max} > \mu_{\min} > 0$, and the $\beta$-margin condition holds (along with some other assumptions), an excess risk of $n^{-\alpha_H(\beta+1)/(2\alpha_H+d)}$ is achievable and is also the best possible. This is exactly the rate we obtain for nearest neighbor classification, once we translate between the different notions of smoothness as per Lemma 2.

We discuss other interesting scenarios in Section C.4 in the appendix.

## 2.3   A general upper bound on the misclassification error

We now get to our most general finite-sample bound. It requires no assumptions beyond the basic measurability conditions stated at the beginning of Section 2, and it is the basis of the all the results described so far. We begin with some key definitions.

**The radius and probability-radius of a ball.**  When dealing with balls, we will primarily be interested in their probability mass. To this end, for any $x \in \mathcal{X}$ and any $0 \leq p \leq 1$, define

$$r_p(x) = \inf\{r \mid \mu(B(x,r)) \geq p\}.$$

Thus $\mu(B(x, r_p(x))) \geq p$ (Lemma 23), and $r_p(x)$ is the smallest radius for which this holds.

**The effective interiors of the two classes, and the effective boundary.**  When asked to make a prediction at point $x$, the $k$-NN classifier finds the $k$ nearest neighbors, which can be expected to lie in $B(x, r_p(x))$ for $p \approx k/n$. It then takes an average over these $k$ labels, which has a standard deviation of $\Delta \approx 1/\sqrt{k}$. With this in mind, there is a natural definition for the *effective interior* of the $Y = 1$ region: the points $x$ with $\eta(x) > 1/2$ on which the $k$-NN classifier is likely to be correct:

$$\mathcal{X}_{p,\Delta}^+ = \{x \in \text{supp}(\mu) \mid \eta(x) > \frac{1}{2}, \ \eta(B(x,r)) \geq \frac{1}{2} + \Delta \ \text{ for all } r \leq r_p(x)\}.$$

The corresponding definition for the $Y = 0$ region is

$$\mathcal{X}_{p,\Delta}^- = \{x \in \text{supp}(\mu) \mid \eta(x) < \frac{1}{2}, \ \eta(B(x,r)) \leq \frac{1}{2} - \Delta \ \text{ for all } r \leq r_p(x)\}.$$

The remainder of $\mathcal{X}$ is the effective boundary,

$$\partial_{p,\Delta} = \mathcal{X} \setminus (\mathcal{X}_{p,\Delta}^+ \cup \mathcal{X}_{p,\Delta}^-).$$

Observe that $\partial_{p',\Delta'} \subset \partial_{p,\Delta}$ whenever $p' \leq p$ and $\Delta' \leq \Delta$. Under mild conditions, as $p$ and $\Delta$ tend to zero, the effective boundary tends to the actual decision boundary $\{x \mid \eta(x) = 1/2\}$ (Lemma 14), which we shall denote $\partial_o$.

The misclassification rate of the $k$-NN classifier can be bounded by the probability mass of the effective boundary:

**Theorem 5.** *Pick any $0 < \delta < 1$ and positive integers $k < n$. Let $g_{n,k}$ denote the $k$-NN classifier based on $n$ training points, and $g(x)$ the Bayes-optimal classifier. With probability at least $1 - \delta$ over the choice of training data,*

$$\Pr_X(g_{n,k}(X) \neq g(X)) \ \leq \ \delta + \mu(\partial_{p,\Delta}),$$

*where*

$$p = \frac{k}{n} \cdot \frac{1}{1 - \sqrt{(4/k)\ln(2/\delta)}}, \quad \text{and} \quad \Delta = \min\left(\frac{1}{2}, \sqrt{\frac{1}{k}\ln\frac{2}{\delta}}\right).$$

## 2.4 A general lower bound on the misclassification error

Finally, we give a counterpart to Theorem 5 that lower-bounds the expected probability of error of $g_{n,k}$. For any positive integers $k < n$, we identify a region close to the decision boundary in which a $k$-NN classifier has a constant probability of making a mistake. This high-error set is $\mathcal{E}_{n,k} = \mathcal{E}_{n,k}^+ \cup \mathcal{E}_{n,k}^-$, where

$$\mathcal{E}_{n,k}^+ = \left\{ x \in \text{supp}(\mu) \mid \eta(x) > \frac{1}{2}, \; \eta(B(x,r)) \leq \frac{1}{2} + \frac{1}{\sqrt{k}} \text{ for all } r_{k/n}(x) \leq r \leq r_{(k+\sqrt{k}+1)/n}(x) \right\}$$

$$\mathcal{E}_{n,k}^- = \left\{ x \in \text{supp}(\mu) \mid \eta(x) < \frac{1}{2}, \; \eta(B(x,r)) \geq \frac{1}{2} - \frac{1}{\sqrt{k}} \text{ for all } r_{k/n}(x) \leq r \leq r_{(k+\sqrt{k}+1)/n}(x) \right\}.$$

(Recall the definition (3) of $\eta(A)$ for sets $A$.) For smooth $\eta$ this region turns out to be comparable to the effective decision boundary $\partial_{k/n,1/\sqrt{k}}$. Meanwhile, here is a lower bound that applies to any $(\mathcal{X}, \rho, \mu)$.

**Theorem 6.** *For any positive integers $k < n$, let $g_{n,k}$ denote the $k$-NN classifier based on $n$ training points. There is an absolute constant $c_o$ such that the expected misclassification rate satisfies*

$$\mathbb{E}_n \text{Pr}_X(g_{n,k}(X) \neq g(X)) \geq c_o \, \mu(\mathcal{E}_{n,k}),$$

*where $\mathbb{E}_n$ is expectation over the choice of training set.*

## Acknowledgements

The authors are grateful to the National Science Foundation for support under grant IIS-1162581.

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
