[Supplementary Material]

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

# A Proofs of the general upper and lower bounds (Theorems 5 and 6)

## A.1 A tie-breaking mechanism

In some situations, such as discrete instance spaces, there is a non-zero probability that two or more of the training points will be equidistant from the query point. In practice, we break ties by a simple rule such as preferring points that appear earlier in the sequence. To accurately reflect this in the analysis, we adopt the following mechanism: for each training point $X$, we also draw a value $Z$ independently and uniformly at random from $[0, 1]$. When breaking ties, points with lower $Z$ value are preferred. We use the notation $X' = (X, Z) \in \mathcal{X} \times [0, 1]$ to refer to the *augmented* training instances, drawn from the product measure $\mu \times \nu$, where $\nu$ is Lebesgue measure on $[0, 1]$.

Given a query point $x \in \mathcal{X}$ and training points $X'_1, \ldots, X'_n \in \mathcal{X} \times [0, 1]$, let $X'_{(1)}(x), \ldots, X'_{(n)}(x)$ denote a reordering of these points by increasing distance from $x$, where each $X'_{(i)}$ is of the form $(X_{(i)}, Z_{(i)})$. With probability 1, this ordering is unambiguous. Also, let $Y_{(1)}(x), \ldots, Y_{(n)}(x)$ be the corresponding labels.

We will need to consider balls in the augmented space. For $x_o \in \mathcal{X}$, $r_o \geq 0$, and $z_o \in [0, 1]$, define

$$B'(x_o, r_o, z_o) = \{(x, z) \in \mathcal{X} \times [0, 1] \mid \text{either } \rho(x_o, x) < r_o \text{ or } (\rho(x_o, x) = r_o \text{ and } z < z_o)\}$$

$$= \left(B^o(x_o, r_o) \times [0, 1]\right) \bigcup \left((B(x_o, r_o) \setminus B^o(x_o, r_o)) \times [0, z_o)\right).$$

Given a set of training points $(X'_i, Y_i)$ and an augmented ball $B' = B'(x_o, r_o, z_o)$, let $\widehat{Y}(B')$ denote the mean of the $Y_i$ for points $X'_i \in B'$; if there is no $X'_i \in B'$, then this is undefined.

Let $\eta(B')$ denote the mean probability that $Y = 1$ for points $(x, z) \in B'$; formally, it is given by

$$\eta(B') = \frac{1}{(\mu \times \nu)(B')} \int_{B'} \eta \, d(\mu \times \nu)$$

whenever $(\mu \times \nu)(B') > 0$. Here $\eta(x, z)$ is defined to be $\eta(x)$.

The ball $B' = B(x_o, r_o, z_o)$ in the augmented space can be thought of as lying between the open ball $B^o = B^o(x_o, r_o)$ and the closed ball $B = B(x_o, r_o)$ in the original space; and indeed $\eta(B')$ is a convex combination of $\eta(B)$ and $\eta(B^o)$ (Lemma 25).

## A.2 Proof of Theorem 5

Theorem 5 rests on the following basic observation.

**Lemma 7.** *Let $g_{n,k}$ denote the $k$-NN classifier based on training data $(X'_1, Y_1), \ldots, (X'_n, Y_n)$. Pick any $x_o \in \mathcal{X}$ and any $0 \leq p \leq 1$, $0 \leq \Delta \leq 1/2$. Let $B' = B'(x_o, \rho(x_o, X_{(k+1)}(x_o)), Z_{(k+1)})$. Then*

$$
\begin{aligned}
1(g_{n,k}(x_o) \neq g(x_o)) \quad \leq \quad & 1(x_o \in \partial_{p,\Delta}) + \\
& 1(\rho(x_o, X_{(k+1)}(x_o)) > r_p(x_o)) + \\
& 1(|\widehat{Y}(B') - \eta(B')| \geq \Delta).
\end{aligned}
$$

*Proof.* Suppose $x_o \notin \partial_{p,\Delta}$. Then, without loss of generality, $x_o$ lies in $\mathcal{X}^+_{p,\Delta}$, whereupon $\eta(B(x_o, r)) \geq 1/2 + \Delta$ for all $r \leq r_p(x_o)$.

Next, suppose $r = \rho(x_o, X_{(k+1)}(x_o)) \leq r_p(x_o)$. Then $\eta(B(x_o, r))$ and $\eta(B^o(x_o, r))$ are both $\geq 1/2 + \Delta$ (Lemma 26). By Lemma 25, $\eta(B')$ is a convex combination of these and is thus also $\geq 1/2 + \Delta$.

The prediction $g_{n,k}(x_o)$ is based on the average of the $Y_i$ values of the $k$ points closest to $x_o$, in other words, $\widehat{Y}(B')$. If this average differs from $\eta(B')$ by less than $\Delta$, then it is $> 1/2$, whereupon the prediction is correct. $\qquad \square$

When we take expectations in the inequality of Lemma 7, we see that there are three probabilities to be bounded. The second of these, the probability that $\rho(x_o, X_{(k+1)}(x_o)) > r_p(x_o)$, can easily be controlled when $p$ is sufficiently large.

**Lemma 8.** *Fix any $x_o \in \mathcal{X}$ and $0 \leq p, \gamma \leq 1$. Pick any positive integer $k \leq (1 - \gamma)np$. Let $X_1, \ldots, X_n$ be chosen uniformly at random from $\mu$. Then*

$$\Pr_n(\rho(x_o, X_{(k+1)}(x_o)) > r_p(x_o)) \leq e^{-np\gamma^2/2} \leq e^{-k\gamma^2/2}.$$

*Proof.* The probability that any given $X_i$ falls in $B(x_o, r_p(x_o))$ is at least $p$ (Lemma 23). The probability that $\leq k \leq (1 - \gamma)np$ of them land in this ball is, by the multiplicative Chernoff bound, at most $e^{-np\gamma^2/2}$. $\square$

To bound the probability that $\widehat{Y}(B')$ differs substantially from $\eta(B')$, a slightly more careful argument is needed.

**Lemma 9.** *Fix any $x_o \in \mathcal{X}$ and any $0 \leq \Delta \leq 1/2$. Draw $(X_1, Z_1, Y_1), \ldots, (X_n, Z_n, Y_n)$ independently at random and let $B' = B'(x_o, \rho(x_o, X_{(k+1)}(x_o)), Z_{(k+1)}) \subset \mathcal{X} \times [0, 1]$. Then*

$$\Pr_n(|\widehat{Y}(B') - \eta(B')| \geq \Delta) \leq 2e^{-2k\Delta^2}.$$

*Moreover, if $\eta(B') \in \{0, 1\}$ then $\widehat{Y}(B') = \eta(B')$ with probability one.*

*Proof.* We will pick the points $X'_i = (X_i, Z_i)$ and their labels $Y_i$ in the following manner:

1. First pick a point $(X_1, Z_1) \in \mathcal{X} \times [0, 1]$ according to the marginal distribution of the $(k + 1)$st nearest neighbor of $x_o$.

2. Pick $k$ points uniformly at random from the distribution $\mu \times \nu$ restricted to $B' = B'(x_o, \rho(x_o, X_1), Z_1)$.

3. Pick $n - k - 1$ points uniformly at random from the distribution $\mu \times \nu$ restricted to $(\mathcal{X} \times [0, 1]) \setminus B'$.

4. Randomly permute the $n$ points obtained in this way.

5. For each $(X_i, Z_i)$ in the permuted order, pick a label $Y_i$ from the conditional distribution $\eta(X_i)$.

The $k$ nearest neighbors of $x_o$ are the points picked in step 2. Their $Y$ values are independent and identically distributed with expectation $\eta(B')$. The main bound in the lemma now follows from a direct application of Hoeffding's inequality.

The final statement of the lemma is trivial and is needed to cover situations in which $\Delta = 1/2$. $\square$

We now complete the proof of Theorem 5. Adopt the settings of $p$ and $\Delta$ from the theorem statement, and define the central bad event to be

$$\text{BAD}(X_o, X'_1, \ldots, X'_n, Y_1, \ldots, Y_n) = \mathbb{1}(\rho(X_o, X_{(k+1)}(X_o)) > r_p(X_o)) + \mathbb{1}(|\widehat{Y}(B') - \eta(B')| \geq \Delta),$$

where $B'$ is a shorthand for $B'(X_o, \rho(X_o, X_{(k+1)}(X_o)), Z_{(k+1)})$, as before. Fix any $x_o \in \mathcal{X}$. If $\Delta < 1/2$, then by Lemmas 8 and 9,

$$\mathbb{E}_n \text{BAD}(x_o, X'_1, \ldots, X'_n, Y_1, \ldots, Y_n) \leq \exp(-k\gamma^2/2) + 2\exp(-2k\Delta^2) \leq \delta^2,$$

where $\gamma = 1 - (k/np) = \sqrt{(4/k)\ln(2/\delta)}$ and $\mathbb{E}_n$ is expectation over the choice of training data. If $\Delta = 1/2$ then $\eta(B') \in \{0, 1\}$ and we have

$$\mathbb{E}_n \text{BAD}(x_o, X'_1, \ldots, X'_n, Y_1, \ldots, Y_n) \leq \exp(-k\gamma^2/2) \leq \delta^2.$$

Taking expectation over $X_o$,

$$\mathbb{E}_{X_o} \mathbb{E}_n \text{BAD}(X_o, X'_1, \ldots, X'_n, Y_1, \ldots, Y_n) \leq \delta^2,$$

from which, by switching expectations and applying Markov's inequality, we have

$$\Pr_n(\mathbb{E}_{X_o} \text{BAD}(X_o, X'_1, \ldots, X'_n, Y_1, \ldots, Y_n) \geq \delta) \leq \delta.$$

The theorem then follows by writing the result of Lemma 7 as

$$\Pr_{X_o}(g_{n,k}(X_o) \neq g(X_o)) \leq \mu(\partial_{p,\Delta}) + \mathbb{E}_{X_o} \text{BAD}(X_o, X'_1, \ldots, X'_n, Y_1, \ldots, Y_n).$$

## A.3 Proof of Theorem 6

For positive integer $n$ and $0 \leq p \leq 1$, let $\text{bin}(n, p)$ denote the (binomial) distribution of the sum of $n$ independent Bernoulli($p$) random variables. We will use $\text{bin}(n, p; \geq k)$ to denote the probability that this sum is $\geq k$; and likewise $\text{bin}(n, p; \leq k)$.

It is well-known that the binomial distribution can be approximated by a normal distribution, suitably scaled. Slud [14] has finite-sample results of this form that will be useful to us.

**Lemma 10.** *Pick any $0 < p \leq 1/2$ and any nonnegative integer $\ell$.*

    *(a) [14, p. 404, item (v)] If $\ell \leq np$, then $\text{bin}(n, p; \geq \ell) \geq 1 - \Phi((\ell - np)/\sqrt{np})$.*

    *(b) [14, Thm 2.1] If $np \leq \ell \leq n(1 - p)$, then $\text{bin}(n, p; \geq \ell) \geq 1 - \Phi((\ell - np)/\sqrt{np(1 - p)})$.*

*Here $\Phi(a) = (2\pi)^{-1/2} \int_{-\infty}^{a} \exp(-t^2/2)dt$ is the cumulative distribution function of the standard normal.*

Now we begin the proof of Theorem 6. Fix any integers $k < n$, and any $x_o \in \mathcal{E}_{n,k}$. Without loss of generality, $\eta(x_o) < 1/2$.

Pick $X_1, \ldots, X_n$ and $Z_1, \ldots, Z_n$ (recall the discussion on tie-breaking in Section A.1) in the following manner:

1. First pick a point $(X_1, Z_1) \in \mathcal{X} \times [0, 1]$ according to the marginal distribution of the $(k + 1)$st nearest neighbor of $x_o$.

2. Pick $k$ points uniformly at random from the distribution $\mu \times \nu$ restricted to $B' = B'(x_o, \rho(x_o, X_1), Z_1)$; recall the earlier definition of the augmented space $\mathcal{X} \times [0, 1]$ and augmented balls within this space.

3. Pick $n - k - 1$ points uniformly at random from the distribution $\mu \times \nu$ restricted to $(\mathcal{X} \times [0, 1]) \setminus B'$.

4. Randomly permute the $n$ points obtained in this way.

The $(k + 1)$st nearest neighbor of $x_o$, denoted $X_{(k+1)}(x_o)$, is the point chosen in the first step. With constant probability, it lies within a ball of probability mass $(k + \sqrt{k} + 1)/n$ centered at $x_o$, but not within a ball of probability mass $k/n$. Call this event $G_1$:

$$G_1: \quad r_{k/n}(x_o) \leq \rho(x_o, X_{(k+1)}(x_o)) \leq r_{(k+\sqrt{k}+1)/n}(x_o)$$

**Lemma 11.** *There is an absolute constant $c_1 > 0$ such that $\Pr(G_1) \geq c_1$.*

*Proof.* The expected number of points $X_i$ that fall in $B(x_o, r_{(k+\sqrt{k}+1)/n}(x_o))$ is $\geq k + \sqrt{k} + 1$; the probability that the actual number is $\leq k$ is at most $\text{bin}(n, (k + \sqrt{k} + 1)/n; \leq k)$. Likewise, the expected number of points that fall in $B^o(x_o, r_{k/n}(x_o))$ is $\leq k$, and the probability that the actual number is $\geq k + 1$ is at most $\text{bin}(n, k/n; \geq k + 1)$. If neither of these bad events occurs, then $G_1$ holds. Therefore,

$$\Pr(G_1) \geq 1 - \text{bin}\left(n, \frac{k + \sqrt{k} + 1}{n}; \leq k\right) - \text{bin}\left(n, \frac{k}{n}; \geq k + 1\right).$$

The last term is easy to bound: it is $\leq 1/2$ since $k$ is the median of $\text{bin}(n, k/n)$ [11]. To bound the first term, we use Lemma 10(a):

$$\text{bin}\left(n, \frac{k + \sqrt{k} + 1}{n}; \leq k\right) = 1 - \text{bin}\left(n, \frac{k + \sqrt{k} + 1}{n}; \geq k + 1\right)$$

$$\leq \Phi\left(\frac{(k + 1) - (k + \sqrt{k} + 1)}{\sqrt{k + \sqrt{k} + 1}}\right) \leq \Phi(-1/\sqrt{3}),$$

which is $1/2 - c_1$ for some constant $c_1 > 0$. Thus $\Pr(G_1) \geq c_1$. $\qquad\square$

Next, we lower-bound the probability that (conditional on event $G_1$), the $k$ nearest neighbors of $x_o$ have an average $Y$ value with the wrong sign. Recalling that $\eta(x_o) < 1/2$, define the event

$$G_2 : \quad \widehat{Y}(B') > 1/2$$

where as before, $B'$ denotes the ball $B'(x_o, X_{(k+1)}(x_o), Z_{k+1})$ in the augmented space.

**Lemma 12.** *There is an absolute constant $c_2 > 0$ such that $\Pr(G_2|G_1) \geq c_2$.*

*Proof.* Event $G_1$ depends only on step 1 of the sampling process. Assuming this event occurs, step 2 consists in drawing $k$ points from the distribution $\mu \times \nu$ restricted to $B'$. Since $x_o \in \mathcal{E}_{n,k}$, we have (by an application of Lemmas 25 and 26) that $\eta(B') \geq 1/2 - 1/\sqrt{k}$. Now, $\widehat{Y}(B')$ follows a $\text{bin}(k, \eta(B'))$ distribution, and hence, by Lemma 10(b),

$$\Pr\left(\widehat{Y}(B') > \frac{k}{2}\right) = \Pr\left(\widehat{Y}(B') \geq \left\lceil \frac{k+1}{2} \right\rceil\right) \geq \Pr\left(Z \geq \frac{2\sqrt{k}+2}{\sqrt{k}}\right)$$

where $Z$ is a standard normal. The last tail probability is at least some constant $c_2$. $\qquad\square$

In summary, for $x_o \in \mathcal{E}_{n,k}$,

$$\Pr_n(g_{n,k}(x_o) \neq g(x_o)) \geq \Pr(G_1 \wedge G_2) \geq c_1 c_2.$$

Taking expectation over $x_o$, we then get

$$\mathbb{E}_n \Pr_X(g_{n,k}(x) \neq g(x)) \geq c_1 c_2 \mu(\mathcal{E}_{n,k}),$$

as claimed.

# B  Proof of universal consistency (Theorem 1)

Recall that we define $R_n = \Pr_X(g_{n,k_n}(X) \neq Y)$. From equation (1), we have:

$$R_n - R^* \leq \Pr_X(\eta(X) \neq 1/2 \text{ and } g_{n,k_n}(X) \neq g(X)).$$

Defining $\partial_o = \{x \in \mathcal{X} \mid \eta(x) = 1/2\}$ to be the decision boundary, we then have the following corollary of Theorem 5.

**Corollary 13.** *Let $(\delta_n)$ be any sequence of positive reals, and $(k_n)$ any sequence of positive integers. For each $n$, define $(p_n)$ and $(\Delta_n)$ as in Theorem 5. Then*

$$\Pr_n\left(R_n - R^* > \delta_n + \mu\left(\partial_{p_n, \Delta_n} \setminus \partial_o\right)\right) \leq \delta_n,$$

*where $\Pr_n$ is probability over the choice of training data.*

For the rest of the proof, assume that $(\mathcal{X}, \rho, \mu)$ satisfies Lebesgue's differentiation theorem: that is, for any bounded measurable $f : \mathcal{X} \to \mathbb{R}$,

$$\lim_{r \downarrow 0} \frac{1}{\mu(B(x,r))} \int_{B(x,r)} f \, d\mu = f(x)$$

for almost all ($\mu$-a.e.) $x \in \mathcal{X}$. We'll see that, as a result, $\mu(\partial_{p_n, \Delta_n} \setminus \partial_o) \to 0$.

**Lemma 14.** *There exists $\mathcal{X}_o \subset \mathcal{X}$ with $\mu(\mathcal{X}_o) = 0$, such that any $x \in \mathcal{X} \setminus \mathcal{X}_o$ with $\eta(x) \neq 1/2$ lies in $\mathcal{X}_{p,\Delta}^+ \cup \mathcal{X}_{p,\Delta}^-$ for some $p, \Delta > 0$.*

*Proof.* As a result of the differentiation condition,

$$\lim_{r \downarrow 0} \eta(B(x,r)) = \lim_{r \downarrow 0} \frac{1}{\mu(B(x,r))} \int_{B(x,r)} \eta \, d\mu = \eta(x) \qquad (4)$$

for almost all ($\mu$-a.e.) $x \in \mathcal{X}$. Let $\mathcal{X}_o$ denote the set of $x$'s for which (4) fails to hold or that are outside $\text{supp}(\mu)$. Then, $\mu(\mathcal{X}_o) = 0$.

Now pick any $x \notin \mathcal{X}_o$ such that $\eta(x) \neq 1/2$. Without loss of generality, $\eta(x) > 1/2$. Set $\Delta = (\eta(x) - 1/2)/2 > 0$. By (4), there is some $r_o > 0$ such that $\eta(B(x,r)) \geq 1/2 + \Delta$ whenever $0 \leq r \leq r_o$. Define $p = \mu(B(x, r_o)) > 0$. Then $r_p(x) \leq r_o$ and $x \in \mathcal{X}_{p,\Delta}^+$. $\qquad\square$

**Lemma 15.** *If $p_n, \Delta_n \downarrow 0$, then*
$$\lim_{n \to \infty} \mu\big(\partial_{p_n, \Delta_n} \setminus \partial_o\big) = 0.$$

*Proof.* Let $A_n = \partial_{p_n, \Delta_n} \setminus \partial_o$. Then $A_1 \supset A_2 \supset A_3 \supset \cdots$. We've seen earlier that for any $x \in \mathcal{X} \setminus (\mathcal{X}_o \cup \partial_o)$ (where $\mathcal{X}_o$ is defined in Lemma 14), there exist $p, \Delta > 0$ such that $x \notin \partial_{p, \Delta}$. Therefore,
$$\bigcap_{n \geq 1} A_n \subset \mathcal{X}_o,$$
whereupon, by continuity from above, $\mu(A_n) \to 0$. $\qquad\square$

Convergence in probability follows immediately.

**Lemma 16.** *If $k_n \to \infty$ and $k_n/n \to 0$, then for any $\epsilon > 0$,*
$$\lim_{n \to \infty} \Pr_n(R_n - R^* > \epsilon) = 0.$$

*Proof.* First define $\delta_n = \exp(-k_n^{1/2})$, and define the corresponding $p_n, \Delta_n$ as in Theorem 5. It is easily checked that the three sequences $\delta_n, p_n, \Delta_n$ all go to zero.

Pick any $\epsilon > 0$. By Lemma 15, we can choose a positive integer $N$ so that $\delta_n \leq \epsilon/2$ and $\mu(\partial_{p_n, \Delta_n} \setminus \partial_o) \leq \epsilon/2$ whenever $n \geq N$. Then by Corollary 13, for $n \geq N$,
$$\Pr_n(R_n - R^* > \epsilon) \leq \delta_n.$$
Now take $n \to \infty$. $\qquad\square$

We finish with almost sure convergence.

**Lemma 17.** *Suppose that in addition to the conditions of Lemma 16, we have $k_n/(\log n) \to \infty$. Then $R_n \to R^*$ almost surely.*

*Proof.* Choose $\delta_n = 1/n^2$, and for each $n$ set $p_n, \Delta_n$ as in Theorem 5. It can be checked that the resulting sequences $(p_n)$ and $(\Delta_n)$ both go to zero.

Pick any $\epsilon > 0$. Choose $N$ so that $\sum_{n \geq N} \delta_n \leq \epsilon$. Letting $\omega$ denote a realization of an infinite training sequence $(X_1, Y_1), (X_2, Y_2), \ldots$, we have from Corollary 13 that
$$\Pr\big\{\omega \mid \exists n \geq N : R_n(\omega) - R^* > \delta_n + \mu\big(\partial_{p_n, \Delta_n} \setminus \partial_o\big)\big\} \leq \sum_{n \geq N} \delta_n \leq \epsilon.$$

Therefore, with probability at least $1 - \epsilon$ over $\omega$, we have
$$R_n(\omega) - R^* \leq \delta_n + \mu\big(\partial_{p_n, \Delta_n} \setminus \partial_o\big)$$
for all $n \geq N$, whereupon, by Lemma 15, $R_n(\omega) \to R^*$. The result follows since this is true for any $\epsilon > 0$. $\qquad\square$

## C  Generalization bounds under smoothness conditions

### C.1  Proof of Lemma 2

Suppose that $\eta$ satisfies the $\alpha$-Holder condition so that for some constant $C > 0$,
$$|\eta(x) - \eta(x')| \leq C\|x - x'\|^{\alpha_H}$$
whenever $x, x' \in \mathcal{X}$. For any $x \in \text{supp}(\mu)$ and $r > 0$, we then have
$$|\eta(x) - \eta(B(x, r))| \leq Cr^{\alpha_H}.$$
If $\mu$ has a density that is lower-bounded by $\mu_{\min}$, and $B(x, r) \subset \mathcal{X}$, we also have
$$\mu(B^o(x, r)) \geq \mu_{\min} v_d r^d,$$
where $v_d$ is the volume of the unit ball in $\mathbb{R}^d$. The lemma follows by combining these two inequalities.

## C.2 Proof of Theorem 3

Recall that the key term in the upper bound (Theorem 5) is $\mu(\partial_{p,\Delta})$, for $p \approx k/n$ and $\Delta \approx 1/\sqrt{k}$.

**Lemma 18.** *If $\eta$ is $(\alpha, L)$-smooth in $(\mathcal{X}, \rho, \mu)$, then for any $p, \Delta \geq 0$,*

$$\partial_{p,\Delta} \cap \operatorname{supp}(\mu) \ \subset \ \left\{ x \in \mathcal{X} \ \middle| \ \left| \eta(x) - \frac{1}{2} \right| \leq \Delta + Lp^{\alpha} \right\}.$$

*Proof.* Pick any $x \in \operatorname{supp}(\mu)$ and any $p \geq 0$. For $r \leq r_p(x)$, we have $\mu(B^o(x,r)) \leq p$ and thus, by the definition of $(\alpha, L)$-smoothness,

$$|\eta(B(x,r)) - \eta(x)| \ \leq \ Lp^{\alpha}.$$

As a result, if $\eta(x) > 1/2 + \Delta + Lp^{\alpha}$ then $\eta(B(x,r)) > 1/2 + \Delta$ whenever $r \leq r_p(x)$. Therefore, such an $x$ lies in the effective interior $\mathcal{X}^+_{p,\Delta}$. A similar result applies to $x$ with $\eta(x) < 1/2 - \Delta - Lp^{\alpha}$. Therefore, the boundary region $\partial_{p,\Delta}$ can only contain points $x$ for which $|\eta(x) - 1/2| \leq \Delta + Lp^{\alpha}$, as claimed. $\square$

This yields a bound on $\operatorname{Pr}_X(g_{n,k}(X) \neq g(X))$ that is roughly of the form $\mu(\{x \mid |\eta(x) - 1/2| \leq k^{-1/2} + L(k/n)^{\alpha}\})$. The optimal setting of $k$ is then $\sim n^{2\alpha/(2\alpha+1)}$.

The key term in the lower bound of Theorem 6 is $\mu(\mathcal{E}_{n,k})$. Under the smoothness condition, it becomes directly comparable to the upper bound.

**Lemma 19.** *If $\eta$ is $(\alpha, L)$-smooth in $(\mathcal{X}, \rho, \mu)$, then for any $k, n$,*

$$\mathcal{E}_{n,k} \supset \left\{ x \in \operatorname{supp}(\mu) \ \middle| \ \eta(x) \neq \frac{1}{2}, \ \left| \eta(x) - \frac{1}{2} \right| \leq \frac{1}{\sqrt{k}} - L\left( \frac{k + \sqrt{k} + 1}{n} \right)^{\alpha} \right\}.$$

*Proof.* The proof is similar to that of the upper bound. Any point $x \in \operatorname{supp}(\mu)$ with

$$\frac{1}{2} < \eta(x) \leq \frac{1}{2} + \frac{1}{\sqrt{k}} - L\left( \frac{k + \sqrt{k} + 1}{n} \right)^{\alpha}$$

has $\eta(B(x,r)) \leq 1/2 + 1/\sqrt{k}$ for all $r \leq r_{(k+\sqrt{k}+1)/n}(x)$, and therefore lies in $\mathcal{E}^+_{n,k}$. Likewise for $\mathcal{E}^-_{n,k}$. $\square$

## C.3 Proof of Theorem 4

Assume that $\eta$ is $(\alpha, L)$-smooth in $(\mathcal{X}, \rho, \mu)$, that is,

$$|\eta(B(x,r)) - \eta(x)| \leq L\mu(B^o(x,r))^{\alpha} \tag{5}$$

for all $x \in \operatorname{supp}(\mu)$ and all $r > 0$, and also that it satisfies the $\beta$-margin condition (with constant $C$), under which, for any $t \geq 0$,

$$\mu\left( \left\{ x \ \middle| \ \left| \eta(x) - \frac{1}{2} \right| \leq t \right\} \right) \ \leq \ Ct^{\beta}. \tag{6}$$

**Proof of Theorem 4(a)**

Set $p, \Delta$ as specified in Theorem 5. It follows from that theorem and from Lemma 18 that under (5) and (6), for any $\delta > 0$, with probability at least $1 - \delta$ over the choice of training data,

$$\operatorname{Pr}_X(g_{n,k}(X) \neq g(X)) \ \leq \ \delta + \mu(\partial_{p,\Delta}) \ \leq \ \delta + C(\Delta + Lp^{\alpha})^{\beta}.$$

Expanding $p, \Delta$ in terms of $k$ and $n$, this becomes

$$\operatorname{Pr}_X(g_{n,k}(X) \neq g(X)) \ \leq \ \delta + C\left( \left( \frac{\ln(2/\delta)}{k} \right)^{1/2} + L\left( \frac{2k}{n} \right)^{\alpha} \right)^{\beta},$$

provided $k \geq 16 \ln(2/\delta)$. The result follows by setting $k \propto n^{2\alpha/(2\alpha+1)}(\log(1/\delta))^{1/(2\alpha+1)}$.

**Proof of Theorem 4(b)**

Theorem 4(b) is an immediate consequence of Lemma 21 below. We begin, however, with an intermediate result about the pointwise expected risk.

Fix any $n$ and any $k < n$, and set $p = 2k/n$. Define

$$\Delta(x) = |\eta(x) - 1/2|$$
$$\Delta_o = Lp^\alpha$$

Recall that the Bayes classifier $g(x)$ has risk $R^*(x) = \min(\eta(x), 1 - \eta(x))$ at $x$. The pointwise risk of the $k$-NN classifier $g_{n,k}$ is denoted $R_{n,k}(x)$.

**Lemma 20.** *Pick any $x \in \operatorname{supp}(\mu)$ with $\Delta(x) > \Delta_o$. Under (5),*

$$\mathbb{E}_n R_{n,k}(x) - R^*(x) \leq \exp(-k/8) + 4\Delta(x) \exp(-2k(\Delta(x) - \Delta_o)^2).$$

*Proof.* Assume without loss of generality that $\eta(x) > 1/2$. By (5), for any $0 \leq r \leq r_p(x)$, we have

$$\eta(B(x,r)) \geq \eta(x) - Lp^\alpha = \eta(x) - \Delta_o = \frac{1}{2} + (\Delta(x) - \Delta_o),$$

whereby $x \in \mathcal{X}_{p,\Delta(x)-\Delta_o}^+$ (and thus $x \notin \partial_{p,\Delta(x)-\Delta_o}$).

Next, recalling (1), and then applying Lemma 7,

$$R_{n,k}(x) - R^*(x) = 2\Delta(x)1(g_{n,k}(x) \neq g(x))$$
$$\leq 2\Delta(x)\left(1(\rho(x, X_{(k+1)}(x)) > r_p(x)) + 1(|\widehat{Y}(B') - \eta(B')| \geq \Delta(x) - \Delta_o)\right),$$

where $B'$ is as defined in that lemma statement. We can now take expectation over the training data and invoke Lemmas 8 and 9 to conclude

$$\mathbb{E}_n R_{n,k}(x) - R^*(x) \leq 2\Delta(x)\left(\Pr{}_n(\rho(x, X_{(k+1)}(x)) > r_p(x)) + \Pr{}_n(|\widehat{Y}(B') - \eta(B')| \geq \Delta(x) - \Delta_o)\right)$$
$$\leq 2\Delta(x)\left(\exp\left(-\frac{k}{2}\left(1 - \frac{k}{np}\right)^2\right) + 2\exp\left(-2k(\Delta(x) - \Delta_o)^2\right)\right),$$

from which the lemma follows by substituting $p = 2k/n$ and observing $\Delta(x) \leq 1/2$. □

**Lemma 21.** *Under (5) and (6),*

$$\mathbb{E}_n R_{n,k} - R^* \leq \exp(-k/8) + 6C\max\left(2L\left(\frac{2k}{n}\right)^\alpha, \sqrt{\frac{8(\beta + 2)}{k}}\right)^{\beta+1}.$$

*Proof.* Recall the definitions of $p(= 2k/n)$ and $\Delta_o, \Delta(x)$ above. Further, for each integer $i \geq 1$, define $\Delta_i = \Delta_o \cdot 2^i$. Fix any $i_o \geq 1$.

Lemma 20 bounds the expected pointwise risk for any $x$ with $\Delta(x) > \Delta_o$. We will apply it to points with $\Delta(x) > \Delta_{i_o}$. For all remaining $x$, we have $\mathbb{E}_n R_{n,k}(x) - R^*(x) \leq 2\Delta_{i_o}$. Taking expectation over $X$,

$$\mathbb{E}_n R_n - R^*$$
$$\leq \mathbb{E}_X\left[2\Delta_{i_o}1(\Delta(X) \leq \Delta_{i_o}) + \exp(-k/8) + 4\Delta(X)\exp(-2k(\Delta(X) - \Delta_o)^2)1(\Delta(X) > \Delta_{i_o})\right]$$
$$\leq 2\Delta_{i_o}\Pr{}_X(\Delta(X) \leq \Delta_{i_o}) + \exp(-k/8) + 4\mathbb{E}_X\left[\Delta(X)\exp(-2k(\Delta(X) - \Delta_o)^2)1(\Delta(X) > \Delta_{i_o})\right].$$

By the margin condition (6), we have $\Pr_X(\Delta(X) \leq t) \leq Ct^\beta$. Thus only the last expectation remains to be bounded. We do so by considering each interval $\Delta_i < \Delta(X) \leq \Delta_{i+1}$ separately:

$$\mathbb{E}_X\left[\Delta(X)\exp(-2k(\Delta(X) - \Delta_o)^2)1(\Delta_i < \Delta(X) \leq \Delta_{i+1})\right]$$
$$\leq \Delta_{i+1}\exp(-2k(\Delta_i - \Delta_o)^2)\Pr{}_X(\Delta(X) \leq \Delta_{i+1})$$
$$\leq C\Delta_{i+1}^{\beta+1}\exp(-2k(\Delta_i - \Delta_o)^2). \tag{7}$$

If we set

$$i_o \;=\; \max\left(1, \left\lceil \log_2 \sqrt{\frac{2(\beta+2)}{k\Delta_o^2}} \right\rceil \right),$$

then for $i \geq i_o$, the terms (7) are upper-bounded by a geometric series with ratio $1/2$. This is because the ratio of two successive terms can be bounded as

$$
\begin{aligned}
\frac{C\Delta_{i+1}^{\beta+1}\exp(-2k(\Delta_i - \Delta_o)^2)}{C\Delta_i^{\beta+1}\exp(-2k(\Delta_{i-1}-\Delta_o)^2)} &= 2^{\beta+1}\exp(-2k((2^i\Delta_o - \Delta_o)^2 - (2^{i-1}\Delta_o - \Delta_o)^2)) \\
&= 2^{\beta+1}\exp(-2k\Delta_o^2((2^i-1)^2 - (2^{i-1}-1)^2)) \\
&\leq 2^{\beta+1}\exp(-2^{2i-1}k\Delta_o^2) \\
&\leq 2^{\beta+1}\exp(-(\beta+2)) \;\leq\; 1/2.
\end{aligned}
$$

Therefore

$$
\begin{aligned}
\mathbb{E}_X &\left[\Delta(X)\exp(-2k(\Delta(X)-\Delta_o)^2)1(\Delta(X) > \Delta_{i_o})\right] \\
&= \sum_{i \geq i_o}\mathbb{E}_X\left[\Delta(X)\exp(-2k(\Delta(X)-\Delta_o)^2)1(\Delta_i < \Delta(X) \leq \Delta_{i+1})\right] \\
&\leq \sum_{i \geq i_o} C\Delta_{i+1}^{\beta+1}\exp(-2k(\Delta_i - \Delta_o)^2) \\
&\leq C\Delta_{i_o}^{\beta+1}\exp(-2k(\Delta_{i_o-1}-\Delta_o)^2) \;\leq\; C\Delta_{i_o}^{\beta+1}.
\end{aligned}
$$

Putting these together, we have $\mathbb{E}_n R_{n,k} - R^* \;\leq\; 6C\Delta_{i_o}^{\beta+1} + e^{-k/8}$. We finish by substituting $\Delta_{i_o} = 2^{i_o}\Delta_o$. $\qquad\square$

### C.4  Margin bounds under various scenarios of interest

One interesting scenario is when $\eta$ is bounded away from $1/2$, that is, there exists some $\Delta^*$ for which

$$\mu\left(\left\{x \;\Big|\; |\eta(x) - \tfrac{1}{2}| \leq \Delta^*\right\}\right) = 0$$

It follows from Lemma 18 that if $\eta$ is $(\alpha, L)$-smooth in $(\mathcal{X}, \rho, \mu)$, then $\mu(\partial_{p,\Delta}) = 0$ whenever $\Delta + Lp^\alpha \leq \Delta^*$. Invoking Theorem 5 with

$$k = \frac{n}{2}\left(\frac{\Delta^*}{2L}\right)^{1/\alpha}, \quad \delta = 2e^{-k(\Delta^*)^2/4},$$

yields an exponentially fast rate of convergence: $\Pr(g_n(X) \neq g(X)) \leq 2e^{-C_o n}$, where $C_o = (\Delta^*)^{2+1/\alpha}/(8(2L)^{1/\alpha})$.

A final case of interest is when $\eta \in \{0, 1\}$ almost everywhere, so that the Bayes risk $R^*$ is zero: that is, there is no inherent uncertainty in the conditional probability distribution $p(y|x)$.

A useful quantity to consider in this case is the *effective interiors* of the classes as a whole:

$$
\begin{aligned}
\mathcal{X}_p^+ &= \{x \in \text{supp}(\mu) \mid \eta(x) = 1, \eta(B(x,r)) = 1 \text{ for all } r \leq r_p(x)\}. \\
\mathcal{X}_p^- &= \{x \in \text{supp}(\mu) \mid \eta(x) = 0, \eta(B(x,r)) = 0 \text{ for all } r \leq r_p(x)\}.
\end{aligned}
$$

Thus, $\mathcal{X}_p^+ = \mathcal{X}_{p,1/2}^+$, and $\mathcal{X}_p^- = \mathcal{X}_{p,1/2}^-$. The rest of $\mathcal{X}$ is the *effective boundary* between the two classes:

$$\partial_p = \mathcal{X} \setminus (\mathcal{X}_p^+ \cup \mathcal{X}_p^-).$$

Incorporating these two quantities into Theorem 5 yields a bound of the following form.

**Lemma 22.** *Let $\delta$ be any positive real and let $k < n$ be positive integers. With probability $\geq 1 - \delta$ over the choice of the training data, the error of the $k$-nearest neighbor classifier $g_{n,k}$ is bounded as:*

$$\Pr_X(g_{n,k}(X) \neq g(X)) \leq \delta + \mu(\partial_p),$$

*where*

$$p = \frac{k}{n} + \frac{2\log(2/\delta)}{n}\left(1 + \sqrt{1 + \frac{k}{\log(2/\delta)}}\right)$$

*Proof.* The proof is the same as that of Theorem 5, except that the probability of the central bad event is different. We will therefore bound the probabilities of these events.

Observe that under the conditions of the lemma, for any $p$,

$$\partial_{p,\frac{1}{2}} = \partial_p$$

Moreover, for any $x_0 \notin \partial_p$, $\eta(B'(x_0, r_p(x_0)))$ is either 0 or 1; this implies that for all $x \in B'(x_0, r_p(x_0))$ except those in a measure zero subset, $\eta(x)$ is either 0 or 1. Therefore, the probability $\Pr(\hat{Y}(B') \neq \eta(B'))$ is zero.

The rest of the lemma follows from plugging this fact in to the proof of Theorem 5 and some simple algebra. $\square$

In particular, observe that since $p$ increases with increasing $k$, and the dependence on $\Delta$ is removed, the best bounds are achieved at $k = 1$ for:

$$p = \frac{1}{n} + \frac{2(1 + \sqrt{2})\log(2/\delta)}{n}$$

This corroborates the admissibility results of [2], which essentially state that there is no $k > 1$ such that the $k$-nearest neighbor algorithm has equal or better error than the 1-nearest neighbor algorithm against all distributions.

## D  Assorted technical lemmas

**Lemma 23.** *For any $x \in \mathcal{X}$ and $0 \leq p \leq 1$, we have $\mu(B(x, r_p(x))) \geq p$.*

*Proof.* Let $r^* = r_p(x) = \inf\{r \mid \mu(B(x,r)) \geq p\}$. For any $n \geq 1$, let $B_n = B(x, r^* + 1/n)$. Thus $B_1 \supset B_2 \supset B_3 \supset \cdots$, with $\mu(B_n) \geq p$. Since $B(x, r^*) = \bigcap_n B_n$, it follows by continuity from above of probability measures that $\mu(B_n) \to \mu(B(x, r^*))$, so this latter quantity is $\geq p$. $\square$

**Lemma 24** (Cover-Hart). *$\mu(\mathrm{supp}(\mu)) = 1$.*

*Proof.* Let $\mathcal{X}_o$ denote a countable dense subset of $\mathcal{X}$. Now, pick any point $x \notin \mathrm{supp}(\mu)$; then there is some $r > 0$ such that $\mu(B(x, r)) = 0$. It is therefore possible to choose a ball $B_x$ centered in $\mathcal{X}_o$, with rational radius, such that $x \in B_x$ and $\mu(B_x) = 0$. Since there are only countably many balls of this sort,

$$\mu(\mathcal{X} \setminus \mathrm{supp}(\mu)) \leq \mu\Big(\bigcup_{x \notin \mathrm{supp}(\mu)} B_x\Big) = 0.$$

$\square$

**Lemma 25.** *Pick any $x_o \in \mathrm{supp}(\mu)$, $r_o > 0$, and any Borel set $I \subset [0, 1]$. Define $B^o = B^o(x_o, r_o)$ and $B = B(x_o, r_o)$ to be open and closed balls centered at $x_o$, and let $A \subset \mathcal{X} \times [0, 1]$ be given by $A = (B^o \times [0,1]) \bigcup ((B \setminus B^o) \times I)$. Then*

$$\eta(A) = \frac{\mu(B)\nu(I)}{\mu(B)\nu(I) + \mu(B^o)(1 - \nu(I))}\eta(B) + \frac{\mu(B^o)(1 - \nu(I))}{\mu(B)\nu(I) + \mu(B^o)(1 - \nu(I))}\eta(B^o).$$

*Proof.* Since $x_o$ lies in the support of $\mu$, we have $(\mu \times \nu)(A) \geq \mu(B^o) > 0$; hence $\eta(A)$ is well-defined.

$$
\begin{aligned}
\eta(A) &= \frac{1}{(\mu \times \nu)(A)} \int_A \eta \, d(\mu \times \nu) \\
&= \frac{1}{\mu(B^o) + \mu(B \setminus B^o)\nu(I)} \left( \int_{B^o} \eta \, d\mu + \int_{B \setminus B^o} \nu(I)\eta \, d\mu \right) \\
&= \frac{1}{\mu(B^o) + (\mu(B) - \mu(B^o))\nu(I)} \left( \int_{B^o} \eta \, d\mu + \nu(I) \left( \int_B \eta \, d\mu - \int_{B^o} \eta \, d\mu \right) \right) \\
&= \frac{\mu(B^o)\eta(B^o) + \nu(I)(\mu(B)\eta(B) - \mu(B^o)\eta(B^o))}{\mu(B^o)(1 - \nu(I)) + \mu(B)\nu(I)},
\end{aligned}
$$

as claimed. □

**Lemma 26.** *Suppose that for some $x_o \in \operatorname{supp}(\mu)$ and $r_o > 0$ and $q > 0$, it is the case that $\eta(B(x_o, r)) \geq q$ whenever $r \leq r_o$. Then $\eta(B^o(x_o, r_o)) \geq q$ as well.*

*Proof.* Let $B^o = B^o(x_o, r_o)$. Since

$$
B^o = \bigcup_{r < r_o} B(x_o, r),
$$

it follows from the continuity from below of probability measures that

$$
\lim_{r \uparrow r_o} \mu(B(x_o, r)) = \mu(B^o)
$$

and by dominated convergence that

$$
\lim_{r \uparrow r_o} \int_{B(x_o, r)} \eta \, d\mu = \lim_{r \uparrow r_o} \int_{B^o} 1(x \in B(x_o, r))\eta(x)\mu(dx) = \int_{B^o} \eta \, d\mu.
$$

For any $r \leq r_o$, we have $\eta(B(x_o, r)) \geq q$, which can be rewritten as

$$
\int_{B(x_o, r)} \eta \, d\mu - q \, \mu(B(x_o, r)) \geq 0.
$$

Taking the limit $r \uparrow r_o$, we then get the desired statement. □