[Reviews · NeurIPS 2014]

Submitted by Assigned_Reviewer_28

The paper studies convergence rate for k-nearest neighbor classifiers. The authors provide finite sample bounds on the excess risk of these classifiers. When taken to the limit these bounds reproduce the known consistency results for this class. However, they are superior in two ways:
1. They apply in the finite case
2. They apply to a broader set of metric spaces

The presentation is very clear and the intuition is well described. The proofs appear in the appendix and I was able to verify some of them.

The hope that the authors will elaborate more on the way these bounds may be used. Can the quantities appearing in these bounds be empirically estimated? This will make the results more useful.
Summary: A new bounds for the excess risk of k-NN classifiers is derived. Nice presentation, good and solid results that improve on the state of the art.

Submitted by Assigned_Reviewer_29

This paper develops a new way of analyzing k-Nearest Neighbor prediction for classification problems. Nearest Neighbor prediction is a simple and well studied classification method that, although the first results date decades back, has recently seen some renewed interest and new development. This paper provides a new angle on Nearest Neighbor analysis by incorporating that the behavior of NN classifiers actually automatically adapt to local scaling (with respect to the probability distribution) of the input space. This aspect has not been taken into account in previous studies on NN.

Previous analysis of NN has been in terms of smoothness parameters and bounds provided reflect some sort of worst case over the data space of this smoothness parameters. Bound derived like this can be rather loose. This study aims to overcome this drawback. The authors define a new notion of margin (that depends on k and n) and bound the expected error (with repect to the Bayes) in terms of the probability weight of this margin. This is complemented with a lower bound in terms of a similar notion. As a result, the authors obtain a general consistency result as well as some bounds under additional assumptions.

The drawback of the paper is that most of the results have a bit of a “in the middle of the analysis” feel (with the exception of Section 2.6). It is not clear how useful it is to say that the error is bounded by the weight of some subset of the domain, when there is no handle/estimate on how much weight there is on that set...

However, I do think that this work provides sufficient novel insights on the behavior of NN classifiers to justify publication at NIPS. Moreover, the paper is a nice read.
Summary: This paper develops a new way of analyzing k-Nearest Neighbor prediction for classification problems. It contains novel results and techniques on NN classification and will be interesting to the theory sub-community at NIPS.

Submitted by Assigned_Reviewer_40

This paper studies the convergence rate for K-nearest neighbor classifier, showing the difficulty of classification depends on the vicinity of the boundary. The authors derived a upper bound such that the convergence is bounded by this quantity with high probability, and they also derived upper and lower bounds by introducing a Holder-like smoothness condition that is tailored to nearest neighborhood. The notations of this paper are clear and consistent. However, the paper does not give numerical results, nor does it provide enough intuitive interpretations to elucidate the main results.
Summary: The main result of provides the convergence rates for K-nearest neighbor classifier and also gave upper and lower bounds. But it does not have numerical simulations and lacks enough intuitive interpretations.
Author Feedback
Author feedback is not available.